# Neuron–Glia Interactions in Tuberous Sclerosis Complex Affect the Synaptic Balance in 2D and Organoid Cultures

**DOI:** 10.3390/cells10010134

**Published:** 2021-01-12

**Authors:** Stephanie Dooves, Arianne J. H. van Velthoven, Linda G. Suciati, Vivi M. Heine

**Affiliations:** 1Department of Child & Youth Psychiatry, Amsterdam UMC, Amsterdam Neuroscience, Vrije Universiteit Amsterdam, 1081 HV Amsterdam, The Netherlands; s.dooves@amsterdamumc.nl (S.D.); a.vanvelthoven@maastrichtuniversity.nl (A.J.H.v.V.); groenendijk.linda@gmail.com (L.G.S.); 2Centre for Neurogenomics and Cognitive Research, Department of Complex Trait Genetics, Amsterdam Neuroscience, Vrije Universiteit Amsterdam, 1081 HV Amsterdam, The Netherlands

**Keywords:** astrocytes, excitation/inhibition balance, tuberous sclerosis complex, EGF signaling, astrocyte-conditioned medium, iPSC, organoid

## Abstract

Tuberous sclerosis complex (TSC) is a genetic disease affecting the brain. Neurological symptoms like epilepsy and neurodevelopmental issues cause a significant burden on patients. Both neurons and glial cells are affected by TSC mutations. Previous studies have shown changes in the excitation/inhibition balance (E/I balance) in TSC. Astrocytes are known to be important for neuronal development, and astrocytic dysfunction can cause changes in the E/I balance. We hypothesized that astrocytes affect the synaptic balance in TSC. TSC patient-derived stem cells were differentiated into astrocytes, which showed increased proliferation compared to control astrocytes. RNA sequencing revealed changes in gene expression, which were related to epidermal growth factor (EGF) signaling and enriched for genes that coded for secreted or transmembrane proteins. Control neurons were cultured in astrocyte-conditioned medium (ACM) of TSC and control astrocytes. After culture in TSC ACM, neurons showed an altered synaptic balance, with an increase in the percentage of VGAT^+^ synapses. These findings were confirmed in organoids, presenting a spontaneous 3D organization of neurons and glial cells. To conclude, this study shows that TSC astrocytes are affected and secrete factors that alter the synaptic balance. As an altered E/I balance may underlie many of the neurological TSC symptoms, astrocytes may provide new therapeutic targets.

## 1. Introduction

Tuberous sclerosis complex (TSC) is a multisystem disorder characterized by benign tumors, which can present in multiple organ systems. The spectrum of neurological manifestations is broad, although over 90% of the TSC patients has epilepsy and many show TSC-associated neuropsychiatric disorders (TAND), which include neurodevelopmental problems like autism and intellectual disability [1]. TSC is caused by heterozygous mutations in the *TSC1* or *TSC2* genes, which lead to hyperactivation of mTOR signaling. MTOR signaling is involved in intercellular processes like development, proliferation, translation and autophagy [2]. The brains of TSC patients can show cortical malformations (cortical tubers), tumors (SEGAs) and myelin abnormalities [3,4,5]. Analysis of postmortem tissue showed dysplastic neurons in cortical tubers, dysmorphic and reactive astrocytes, microglia activation and hypomyelination [3]. There is urgent need for improved treatment strategies. Up to 50% of TSC patients suffer from seizures refractory to antiepileptic drugs [6], and current medications do not reduce TAND symptoms [7,8], while causing debilitating burdens on patients and families. Although different cell types in the TSC brain are affected, neuronal cell dysfunctions have mainly been considered underlying neural network changes in TSC.

It has been hypothesized that mTOR hyperactivation disrupts the synaptic excitation/inhibition (E/I) balance and thereby causing many of the neurological symptoms of TSC patients [9,10]. Disruptions in the E/I balance in brain networks have been shown to cause epilepsy, but recent data indicates that subtle changes in the E/I balance may also underlie autism [11,12,13]. Previous studies have shown hyperactive neuronal networks in TSC mouse models and stem cell-derived human neuron cultures [9,14,15,16]. These changes in network excitability are often associated with changes in synaptic balance, like a decrease in synaptic inhibition [17] or a decrease in γ-aminobutyric acid (GABA) receptors [9,16]. It is well known that astrocytes play crucial roles in neurodevelopment and can directly influence synaptic transmission [18]. In a mouse model for TSC, astrocytes show a decreased expression of the glutamate transporters GLT1 and GLAST, which led to excitotoxic neuronal cell death and impaired synaptic plasticity [19,20]. Other studies in TSC mutants showed impairments in potassium uptake by astrocytes [21], abnormal gap junction coupling [22], and an increase in astrocytic AQP4 expression [23]. These astrocytic changes can (partly) underlie dysfunctions in neural networks and promote epileptic seizures, showing that the effect of TSC astrocytes on neuronal functioning needs further investigation.

Mouse models for TSC have been used to study brain abnormalities and gave valuable insight into the role of mTOR signaling in the brain. However, TSC mouse models generally carry homozygous knockout mutations of either *Tsc1* or *Tsc2*, which is distinct from patients. Furthermore, rodent astrocytes are markedly different from human astrocytes in terms of size and gene expression [24]. The aim of this study was to test the effect of human TSC patient-derived astrocytes on neuronal network formation. We used induced pluripotent stem cells (iPSCs) of TSC patients and controls. TSC patient iPSC-derived astrocyte lineage cells showed expected pathology like an increased proliferation. Medium conditioned by TSC patient astrocytes altered the synaptic balance in control iPSC-derived neuron cultures. To test whether these phenotypes are also presented in a more physiological, cellular complex model for TSC, we further studied a 3D organoid model containing neurons and glial cells. Organoids confirmed synaptic changes in TSC, showing that TSC astrocytes indeed cause disturbances in synaptic balance and may provide a new target for improved treatments.

## 2. Materials and Methods

### 2.1. iPSC Production

Fibroblasts from four TSC patients and four controls (Table 1) were transfected with lentiviral construct expressing all four Yamanaka factors coupled to a red fluorescent tag [25], as previously described [14]. IPSC lines were maintained on Vitronectin (Stem Cell Technologies, Vancouver, BC, Canada, #7180) coated plates in TeSR™-E8™ medium (Stem Cell Technologies, Vancouver, BC, Canada, #5940). Passaging was done with Gentle Cell Dissociation Reagent (Stem Cell Technologies, Vancouver, BC, Canada, #7174) and gentle tapping to only dissociate pluripotent colonies. For each patient, one iPSC line was used for experiments. All experiments were exempt from approval of the institutional review board of the VU Medical Center.

### 2.2. Astrocyte Differentiation

IPSCs were differentiated into astrocytes according to previously established protocol [26,27]. Briefly, iPSCs were dissociated from the plate with ethylenediaminetetraacetic acid (EDTA; Invitrogen, Waltham, MA, USA, #15575-038) and plated on an antiadherent plate in N2B27 + vit A supplemented with epidermal growth factor (EGF; 20 ng/mL, Peprotech, London, UK, #AF-100-15-500 ug), basic fibroblast growth factor (bFGF; 4 ng/mL, Peprotech, London, UK, #100-18B-50 ug), triiodothyronine (T3; 40 ng/mL, Sigma-Aldrich, Saint Louis, MO, USA, #T6397-100 mg) and rock inhibitor (RI; 10 μM, Selleckchem, Munich, Germany, #S1049) to form embryoid bodies (EBs), with medium refreshment every other day. From day 2 on, the medium was additionally supplemented with retinoic acid (RA; 10 μM, Sigma-Aldrich, Saint Louis, MO, USA, #R2625-100 mg). At day 10, EBs were plated on a Geltrex-coated plate (Life Technologies, Waltham, MA, USA, #A1413302) in N2B27 + vit A supplemented with EGF (20 ng/mL) and T3 (40 ng/mL). If the center of the EBs darkened due to lack of medium infiltration into the core before day 10, EBs were plated earlier (but not before day 4). However, the medium was not changed before day 10. From this moment on until the end of the protocol, cultures were split when confluent by 2–5 min accutase (Merck-Millipore, Darmstadt, Germany, #sf006) incubation followed by tapping the plate to lose cells, spinning the cells down in basal medium and replating on a new Geltrex-coated plate. The medium was refreshed three times a week. At day 18, the medium was changed to N2B27—vit A supplemented with EGF (20 ng/mL) and T3 (40 ng/mL). At day 37, the medium was changed to N2B27—vit A supplemented with EGF (5 ng/mL), T3 (40 ng/mL), bFGF (5 ng/mL), laminin (1 μg/mL, Sigma-Aldrich, Saint Louis, MO, USA, #L2020-1 mg) and vitamin C (50 μg/mL, Sigma-Aldrich, Saint Louis, MO, USA, #A4544-25G). At day 39, the medium was changed to N2B27—vit A supplemented with EGF (5 ng/mL), T3 (40 ng/mL), bFGF (5 ng/mL), laminin (1 μg/mL), vitamin C (50 μg/mL) and Dorsomorphin (0.1 μM, Tocris Bioscience, Bristol, UK, #3093/10). At day 42, the medium was changed to N2B27—vit A supplemented with T3 (40 ng/mL), laminin (1 μg/mL), vitamin C (50 μg/mL) and Dorsomorphin (0.1 μM). At day 45, cells were switched to commercial astrocyte medium (ScienCell Research, Carlsbad, CA, USA, #1801) to promote astrocytic differentiation. At day 60, quality control of astrocyte batches was performed. Cells passed quality control if they presented typical astrocyte morphologies (large nuclei with either a “flat” astrocytic morphology or several large processes), showed GFAP expression by immunostaining and RNA analysis, and expressed additional astrocytic markers, such as CD44, Sox9, ID3, AQP4 and BLBP. None of the astrocyte differentiations failed quality control. After day 60, cells were further matured in commercial astrocyte medium, and passed using accutase treatment upon confluency.

### 2.3. Neuronal Differentiation

Neurons of control lines were differentiated from iPSC according to established protocol [28]. First, iPSCs were differentiated into neuroepithelial stem cells (NES cells) in N2B27 + vit A supplemented with Dorsomorphin (1 μM) and SB431542 (10 μM, Selleckchem, Munich, Germany, #S1067) for dual small mothers against decapentaplegic (SMAD) inhibition. When neural rosettes appeared, they were manually cut and plated onto a new poly-l-ornithine (PLO, Sigma-Aldrich, Saint Louis, MO, USA, #P3655-100 mg)/laminin coated well. The day after, the medium was changed to N2B27 + vit A supplemented with bFGF (20 ng/mL) and EGF (20 ng/mL). Cells were split with TrypLE (Life Technologies, Waltham, MA, USA, #12563-029) when cultures were confluent, and a neuronal induction was started from NES cells at passage 3. To start neuronal induction, the medium was changed to N2 medium supplemented with 400 ng/mL human sonic hedgehog (hSHH, Peprotech, London, UK, #100-45-500ughSHH) when NES cells were confluent. This was considered day 1 of neuronal differentiation. At day 5, the medium was switched to NB medium with 10 μM valproic acid (Sigma-Aldrich, Saint Louis, MO, USA, #P4543-10G). For the first eight days, half of the medium was changed every day. At day 8, cells were split with accutase to a new PLO/laminin coated plate in NB medium supplemented with brain-derived neurotrophic factor (BDNF, 20 ng/mL, Peprotech, London, UK, #450-02), glial cell-derived neurotrophic factors (GDNF, 10 ng/mL, Peprotech, London, UK, #450-010), insulin-like growth factor 1 (IGF1, 10 ng/mL, Peprotech, London, UK, #100-11-100 ug) and cyclic adenosine monophosphate (cAMP, 1 μM, Sigma-Aldrich, Saint Louis, MO, USA, #D0260-5 mg). From day 8 until day 18, half of the medium was refreshed twice a week. At day 18, neurons were frozen down until further use. For the start of an experiment, day 18 neurons were plated on PLO-laminin coated glass coverslips. The day after the neurons were flipped onto a well with rat astrocytes and cultured in ACM 1:1 diluted with fresh NB medium supplemented with cAMP (1 μM). As a control, neurons were cultured on rat astrocytes without ACM. At day 24, neurons were treated with 2 μM Arabinosylcytosine (Merck-Millipore, Darmstadt, Germany #251010). Half of the medium was refreshed twice a week until day 57, when neurons were fixed and used for immunostaining.

### 2.4. Organoid Culture

Organoids were cultured according to an adapted version of Monzel et al. [29]. Briefly, NES cells were produced as described above under “neuronal differentiation”. At passage 4, NES cells were plated on an ultra-low-attachment round-bottom 96-well plate (Corning, Corning, NY, USA, #CLS7007) in a density of 9000 cells per well to form aggregates. Cells were refreshed every other day with N2B27 + vit A supplemented with CHIR99201 (3 μM, Sanbio, Uden, The Netherlands #13122-5), hSHH (400 ng/mL), vitamin C (150 μM) and RI (10 μM). At day 8, the organoids were embedded in matrigel droplets (Corning, Corning, NY, USA, #354277) and cultured in an antiadherent six-well plate with continued medium refreshments every other day. At day 10, the medium was switched to N2B27 + vit A supplemented with BDNF (10 ng/mL), IGF (10 ng/mL), GDNF (10 ng/mL), cAMP (1 μM), vitamin C (150 μM) and T3 (40 ng/mL). Organoids were kept in this medium with feeding every other day until they were collected for analysis on day 30, 60 and 90.

### 2.5. Medium Composition

NB medium: neurobasal medium (Gibco, Waltham, MA, USA, 21103-049) supplemented with 2% B27 with vitamin A (Gibco, Waltham, MA, USA, 17504-044), 18 mM Hepes (Gibco, Waltham, MA, USA, 15630-056), 0.25× Glutamax (Gibco, Waltham, MA, USA, 35050-038), 1% Pen/Strep (Sigma-Aldrich, Saint Louis, MO, USA, P0781).

N2 medium: Dulbecco’s modified Eagle medium/nutrient mixture F12 medium (DMEM/F12) (Life Technologies, Waltham, MA, USA, #21331-046) supplemented with 1% N2 (Thermo Fisher Scientific, Waltham, MA, USA, #17502-048), 1× NEAA (Thermo Fisher Scientific, Waltham, MA, USA, #11140-035), 2 mM L-glutamine (Thermo Fisher Scientific, Waltham, MA, USA, #25030-024), 2 μg/mL heparin (Sigma-Aldrich, Saint Louis, MO, USA, #H3393-50KU), 1% Pen/Strep.

N2B27—vit A: 1:1 DMEM/F12 with neurobasal medium supplemented with 0.5% N2, 1% B27 without vitamin A (Thermo Fisher Scientific, Waltham, MA, USA, #12587-010), 2.5 μg/mL insulin (Sigma-Aldrich, Saint Louis, MO, USA, #I9278), 0.5% L-glutamine, 0.5× NEAA, 50 μM beta-mercaptoethanol (Thermo Fisher Scientific, Waltham, MA, USA, #21985023), 1% Pen/Strep

N2B27 + vit A: 1:1 DMEM/F12 with neurobasal medium supplemented with 0.5% N2, 1% B27 with Vitamin A, 0.25 mg/mL Insulin, 0.5% L-glutamine, 0.5× NEAA, 50 μM beta-mercaptoethanol, 1% Pen/Strep

### 2.6. ACM Collection

To collect astrocyte-conditioned medium (ACM), day-60 astrocytes were plated on two wells of a six-well plate. One of the wells was maintained according to the normal astrocyte culture protocol to expand and maintain astrocytes. The other well was used for ACM collection and switched to 3 mL NB medium the day after plating. After one week, the medium was collected and centrifuged 5 min at 1200 rpm to spin down any cells that might have detached from the astrocyte plate. The supernatant was transferred to another tube, diluted 1:1 with fresh medium and used for medium refreshments of neuron cultures.

### 2.7. Immunostaining

Cells and organoids were fixed with 4% PFA (Electron Microscopy Sciences, Hatfield, PA, USA, #15710-S), with a 15 min RT incubation for cells and a 30 min RT incubation for organoids. Organoids were embedded in Tissue Tek O.C.T. compound (Sakura, Alphen a/d Rijn, The Netherlands) after fixation and cut in 12 µm-thick sections. For immunostaining organoid sections were washed with PBS (137 mM NaCl, 2.7 mM KCl, 10 mM Na_2_HPO_4_, 1.8 mM KH_2_PO_4_) six times in 30 min, and pretreated with warm 0.01 M citrate buffer (0.1 M Citric Acid, 0.1 M Sodium Citrate, pH6.0) for 30 min. Cells were washed with PBS three times in 30 min. After the first wash and citrate pretreatment for organoids the staining protocol was the same for cells and organoids. Slides were incubated in blocking buffer (PBS + 5% goat serum (Life Technologies, Waltham, MA, USA, #16210-064) + 0.1% BSA (Thermo Fisher Scientific, Waltham, MA, USA, #AM2616) + 0.3% TritonX100 (Sigma-Aldrich, Saint Louis, MO, USA, #t8787)) for one hour at RT. Primary antibodies (Table 2) were diluted in blocking buffer and incubated for one hour at RT and then overnight at 4 °C. The next day, slides were washed three times in 30 min with PBS and incubated in secondary antibodies diluted in blocking buffer (Alexa Fluor-488, 568, 594 and 647; 1:1000; Thermo Fisher Scientific, Waltham, MA, USA) for two hours at RT. Slides were washed three times with PBS in 30 min, incubated in 4′,6-diamidino-2-phenylindole (DAPI, 1:1000 in PBS, Sigma-Aldrich, Saint Louis, MO, USA, #D9542) for 2 min, washed with PBS once and embedded with Fluoromount-G (Southern Biotech, Birmingham, AL, USA, #0100-01). Stainings were analyzed on a Leica DM6000B fluorescent microscope (Leica Microsystems, Amsterdam, The Netherlands) for regular stainings, or on a Nikon Eclipse Ti confocal microscope (Nikon Instruments Inc, Amsterdam, The Netherlands) for synapse stainings. To analyze morphology astrocyte plates were scanned on an Opera LX HCS (Perkin Elmer, Waltham, MA, USA).

### 2.8. BrdU Assay

Cells were labelled with 10 μM bromodeoxyuridine (BrdU, Sigma-Aldrich, Saint Louis, MO, USA, B5002) by two-hour incubation at 37 °C. For each line and time point, three conditions were tested: BrdU, BrdU + Rapamycin (30 nM, Sigma-Aldrich, Saint Louis, MO, USA, #553210) and BrdU + EtOH (1:1000; vehicle control for rapamycin). Rapamycin binds mTOR and decreases mTOR activity. After BrdU incubation, cells were washed five times with PBS (2 min each). Cells were fixed with 4% paraformaldehyde (PFA) for 15 min at RT, and washed with PBS for 2 min three times. Cells were incubated in permeabilization buffer (PBS + 0.1% Triton X-100) for 20 min at RT; 10 min in 1 normal solution (N) HCl on ice, 20 min in 2N HCl at RT and 10 min at RT in phosphate/citric acid buffer (182 mM Na_2_HPO_4_, 9 mM citric acid, pH 7.4). After three two-minute washes with permeabilization buffer, cells were incubated in blocking solution for 30 min at RT. Anti-BrdU primary antibody (Table 2) was diluted in blocking buffer and incubated overnight (o/n) at RT. The next day, slides were washed with permeabilization buffer three times for 2 min. Cells were incubated in secondary antibody (anti-rabbit Alexa Fluor 488) diluted in blocking buffer for one hour at RT, after which cells were washed with PBS three times for 2 min, incubated with DAPI (1:1000 in PBS), washed once with PBS and embedded with Fluoromount-G.

### 2.9. RNA Isolation

RNA was isolated from cells and organoids by incubation in 750 μL TRIzol (Thermo Fisher Scientific, Waltham, MA, USA, #15596018). Per 750 μL TRIzol, 150 μL chloroform was added, and samples were shaken vigorously and left to incubate for 2–3 min at RT. Samples were centrifuged at 13.000 *g* for 10 min at 4 °C. The upper aqueous phase was transferred to a new tube and 350 uL isopropanol was added. After 10–15 min incubation at RT, samples were centrifuged at 13.000 *g* for 10 min at 4 °C. The supernatant was removed by decanting, and the pellet is washed with 1 mL 70% EtOH twice, with centrifugation at 7500 *g* for 5 min at 4 °C in between washes. The remaining EtOH was removed with a pipette and the pellet was air dried for 5 min. Samples were dissolved in 10–30 μL DMPC-H2O, and RNA concentration and 260/280 ratios were measured with a NanoDrop spectrophotometer (NanoDrop 2000 Thermo Scientific, Waltham, MA, USA). CDNA was made from 1 μg RNA using Superscript IV (Invitrogen, Waltham, MA, USA, #18090010) and 50 ng/μL random hexamer primers (Qiagen, Germantown, MD, USA, #79236) according to the manufacturer’s protocol.

### 2.10. PCR

For PCR analysis, 1 μL cDNA was mixed with Phire Buffer, dNTPs (Invitrogen, Waltham, MA, USA, #102 97-018), Phire Hot Start II DNA polymerase (Thermo Fisher Scientific, Waltham, MA, USA, #F-122 L) and forward and reverse primers (1 μL each, see Table 2). Samples were run on a thermocycler for 28–32 cycles (5 s 98 °C, 15 s 60 °C and 15 s 72 °C). Afterwards, PCR products were loaded on a 4% agarose gel containing ethidium bromide and run for 45 min at 100 volts. Bands were visualized with UV light. Quantification of band intensity was done with ImageJ software (imagej.nih.gov/ij).

### 2.11. Quantitative PCR

For quantitative analysis, cDNA samples were diluted ten times and run on a Lightcycler 480 (Roche, Woerden, The Netherlands) using a Sensifast Sybr Hi-ROX-kit (Bioline, London, UK, #BIO-92020) according to the manufacturer’s protocol. All samples were run in duplicates or triplicates. Primers targeted housekeeping gene *EIF4G2* and target genes *VGAT*, *VGLUT1* and *NEUN* (Table 3). Samples were divided over two qPCR runs, with 15 samples of the first qPCR repeated in the second qPCR. The ∆∆Ct values were calculated by correction for the housekeeping gene and using the average of the repeated samples in each qPCR as a reference sample, to correct for minor differences between PCR runs. The ∆∆Ct values were calculated into fold changes in gene expression by 2^−∆∆Ct^. The fold changes were averaged between organoids of the same iPSC line and time point, and these averages were used for further statistical analysis.

### 2.12. RNA Sequencing

Total RNA was isolated from three control and four TSC day-60 astrocytes as described under RNA isolation. After Nanodrop measurements, the samples were subsequently measured on an Agilent 2100 Bio Analyzer (Agilent, Santa Clara, CA, USA) to determine RNA integrity number (RIN) scores. All samples had RIN scores higher than 9, so none were excluded for library construction. Illumina^®^ TruSeq Stranded mRNA kit (Illumina, San Diego, CA, USA, #20020594) was used according to manufacturer’s instructions in order to prepare the library, using 150 to 200 ng RNA. The 3′ ends were adenylylated prior to ligation of the adapters to the double-stranded cDNA. In order to enrich the DNA fragments, 15 cycles of PCR were run. The quality of the product was measured on the Agilent D5000 Tape Station (Agilent, Santa Clara, CA, USA) before sequencing.

Samples were aligned to Human Genome hg38. Both expected count and transcripts per kilobase million (TPM) were obtained for in total of 23,369 genes. Genes with TPM < 1 in more than 50% of samples were excluded from the analysis, resulting in 13,239 genes. Differentially expressed gene (DEG) analysis then was performed using DESeq2 with correction for gender. Genes were considered significantly different between TSC and control when the adjusted *p*-value after false discovery rate (FDR) correction was <0.05. Pathway analysis was performed using the protein analysis through evolutionary relationships (PANTHER) overrepresentation test (release 20190711) on all significant DEGs and all genes from the RNA sequencing as a reference list.

### 2.13. Analysis

All experiments were repeated at least twice, and the results of both experiments were averaged to obtain one value per iPSC line. All analyses were either done automated or blinded. Astrocyte and neuronal morphology was analyzed using Columbus software (Perkin Elmer, Waltham, MA, USA). Synaptic analyses were performed in ImageJ (imagej.nih.gov/ij) using the NeuronJ and SynaptoCount plugins. The NeuronJ plugin was used to trace MAP2^+^ dendrites. In SynaptoCount, the dendritic tracing was loaded and used to determine the total number of synapses by counting the number of Synaptophysin1^+^ puncta present on the dendrites. The amount of VGAT^+^ synapses was determined by the number of VGAT^+^/Synaptophysin1^+^ puncta present on dendrites. The synaptic density was calculated by dividing the total number of synapses by the pixel length of the dendrites. The percentage of VGAT^+^ synapses was determined by dividing the number of VGAT^+^ synapses by the total number of synapses. Cell counts in organoids were performed by merging 10× fluorescent images to obtain a composite image of a whole organoid in Adobe Photoshop CS6 (Adobe) using “Photomerge”. Composite images were opened in ImageJ, and each channel was separately converted to a binary image after background subtraction using a rolling ball radius of six pixels. The “analyze particles” function was used to measure cell number. Statistical analysis were done in IBM SPSS statistics software version 26 (IBM). Data was tested for normality with a Shapiro–Wilk test. Significance testing was done with independent-sample *t*-tests (for parametric data) or a Mann–Whitney U test (for nonparametric data).

## 3. Results

### 3.1. TSC Astrocytes Show Increased Proliferation

TSC and control iPSCs were differentiated into astrocytes as previously described [26]. From all lines astrocytes were successfully obtained and no differences in expression of astrocytic markers GFAP, CD44, ID3, SOX9 and Nestin were observed (Figure 1A–C). Astrocytes were analyzed for TSC-associated phenotypes as increased proliferation and altered morphology. No differences in morphology were observed between TSC and control astrocytes at day 60 or day 90 (Figure 1C,D). We performed a proliferation assay on TSC and control astrocytes at different time points during the differentiation. At day 25 (progenitor cell stage) and day 60 (early astrocytes), no significant differences in proliferation were observed (Figure 2A,B). At day 90, TSC astrocytes however showed a significantly higher proliferation rate than control cells (*p* < 0.05), which was rescued by rapamycin treatment (Figure 2C). Rapamycin binds and inhibits mTOR and can thereby rescue some of the lost mTOR inhibition in TSC cells. Cleaved caspase 3 analysis showed no significant differences in apoptosis between TSC and control (Figure 2D). TSC astrocytes show increased proliferation compared to control astrocytes upon maturation, without a change in the rate of apoptosis.

### 3.2. RNA Sequencing Analysis Revealed Changes in Signaling Pathways and Secreted Factors

To further characterize changes between TSC and control astrocytes, we performed RNA sequencing analysis. TSC astrocytes showed 70 differentially expressed genes (DEGs) compared to control astrocytes (Appendix A, Figure 3). Pathway analysis showed enrichment of mTOR-associated processes as “proliferation” and “development” as well as “signaling”, “protein tyrosine phosphatase” and “ECM/membrane proteins” (Figure 3). To test whether these changes in gene expression were consistent over time, a selection of DEGs were tested by PCR in day-60 and day-90 astrocytes (Figure 4A). For *CALB1*, *EFEMP1* and *BAALC,* similar changes were observed in day-60 and day-90 astrocytes, while changes in *BST2* and *ERRFI1* were less pronounced in day-90 astrocytes. Within the signaling category, EGF signaling was the most specific pathway to be enriched in TSC astrocytes. Three EGF signaling genes were upregulated in TSC, *ERRFI1*, *AREG* and *TGFA*. *AREG* and *TGFA* are both ligands for the EGF receptor (EGFR) that are secreted from astrocytes. PCR analysis of EGF signaling components showed no significant changes in day-90 astrocytes, although there is a trend towards increased EGF signaling with a higher mRNA expression for *AREG*, *TGFA*, *EGF* and *INOS* in day-90 astrocytes (Figure 4B). The results of RNA sequencing show 70 DEGs in TSC astrocytes, which are enriched in (EGF) signaling and secreted/transmembrane genes.

### 3.3. TSC Astrocytes Affect Neuronal Development

Based on the RNA sequencing data we decided to study the effect of astrocyte-secreted factors on neuronal maturation and morphology. To test this, we cultured control iPSC-derived neurons in astrocyte-conditioned medium (ACM) from TSC and control astrocytes (Figure 5A). Morphological analysis showed no changes in parameters like dendritic/axonal length, number of neurites and neurite properties like branching or node types (Figure 5B). Although the synaptic density was unchanged in cultures with TSC ACM, the amount of inhibitory (VGAT^+^) synapses was significantly increased (Figure 5C, *p* < 0.05), showing an alteration in the synaptic balance upon culturing in TSC ACM.

### 3.4. Organoids Confirm Alterations in Synaptic Balance

To test changes in the astrocytic cell population in a more physiological model representing cortical neuron–glia interactions, we differentiated TSC patient and control iPSCs into brain organoids. Organoids were collected for RNA and staining at day 30, 60 and 90. For all iPSC lines, RNA was collected from two to five organoids per time point. Staining and RT-PCR analysis indicated presence of neuronal (NeuN, MAP2, SMI312), astrocytic (SOX9, GFAP, CD44, Nestin, ID3), and oligodendrocytic cell lineage cells (OLIG2, CC1, MBP) (Figure 6), indicating presence of mixed neural cell populations. Neurons were observed from the earliest time point on (Figure 6A,B). Oligodendrocyte progenitor cells were present from day 30 onwards, as indicated by presence of OLIG2^+^ and CC1^+^ cells, but mature MBP^+^ oligodendrocytes were only observed at day 90 (Figure 6C). At day 30 a low amount of GFAP^+^ astrocytes were observed, which were increased in number at day 60 and day 90 (Figure 6D–F). No changes in generation of astrocytes (ID3^+^ and SOX9^+^ cells) and neurons (NeuN^+^ cells) or in proliferation (PH3^+^ cells) were observed between TSC and control organoids (Figure 7A,B).

To analyze changes in synaptic balance, we performed qPCR for *NEUN*, *VGAT*, *VGLUT1* and *EIF4G2* (Figure 7C). In TSC patient organoids, *VGLUT1* did not show a gradual increase in expression as shown in control organoids (Figure 7D). *VGLUT1* levels in TSC organoids were decreased at day 60 (D60 *p* = 0.063), which became statistically significant decreased at day 90 (D90 *p* < 0.05), while the expression of *VGAT* remained stable over time. As a consequence, the ratio between *VGAT* and *VGLUT1* expression increased in TSC organoids, although results were not statistically significant due to high variation in the TSC lines (D90 *p* = 0.077). Even though results were not statistically significant, all TSC organoids showed *VGAT*/*VGLUT1* ratios that were higher than the control, but variation was high due to one TSC line that had a strongly increased ratio (Figure 7C). Interestingly, although variation between organoids from different individuals was observed, we did not observe a large variation between organoids from the same iPSC lines (Appendix A), suggestion robustness of the organoid differentiation. Overall, a more physiological model confirmed changes in synaptic balance towards a relatively higher amount of GABAergic synapses in TSC, due to lower numbers of glutamatergic synapses.

## 4. Discussion

TSC patients can suffer from neurological problems like epilepsy or TAND (TSC-associated neuropsychiatric disorders), which cause a high burden for patients. Treatment for TAND is lacking and epilepsy is often refractory, showing the need for new treatment strategies. However, the development of new treatments is hampered by a lack of understanding of the cellular mechanism causing neurological abnormalities. In this study we aimed to contribute to a better understanding of TSC disease mechanisms by studying the effect of TSC astrocytes in neuronal development.

TSC astrocytes showed increased proliferation and changes in gene expression compared to control astrocytes. No changes in morphology were observed, which is consistent with previous studies that showed morphological changes in homozygous TSC knockout cells, but not with heterozygous TSC mutations [30]. Aberrant mTOR activation in TSC is thought to underlie increased proliferation of TSC cells. Indeed, the increased proliferation was rescued by restoration of mTOR inhibition by rapamycin treatment. The DEGs were involved in cellular processes as development, proliferation, signaling and were enriched for secreted and transmembrane proteins. EGF signaling showed up as one of the enriched pathways. To specifically study the effect of astrocyte-secreted factors on neuronal development, we maturated control neurons in ACM of either control or TSC astrocytes. ACM of TSC astrocytes increased the percentage of VGAT^+^ synapses. The increased percentage of VGAT^+^ synapses was unexpected, as this was not previously reported in TSC. To confirm whether this effect is stable in a more physiological setting, i.e., a self-organizing collection of neuronal and glial cells, we decided to test the synaptic balance in organoids. TSC organoids showed a significantly decreased amount of *VGLUT1* mRNA expression, which led to an (nonsignificant) increased *VGAT*/*VGLUT1* balance, suggesting a relative increase in GABAergic synapses like observed in neurons grown in TSC ACM. Overall, the results indicate that TSC astrocytes have an altered secretory profile, which affects neuronal development and changes the synaptic balance.

EGF signaling is involved in neurogenesis by stimulation of proliferation, differentiation and survival of neural stem cells [31]. There is a family of EGF ligands of which EGF, TGFα and AREG exclusively bind the EGF receptor (EGFR). EGFR expression is present throughout the brain and high in neurons. It has been shown that EGF signaling can promote neurite outgrowth and regulate neuronal plasticity by modulating glutamatergic NMDA receptor activation and glutamate release [31]. Therefore, EGF signaling is an interesting candidate for the effects of ACM on neurons observed in this study. EGF and AREG have been identified as potential genetic risk factors for autism [32,33,34]. A number of previous studies showed a role for EGF signaling in TSC. Inhibition of EGF signaling has been shown to decrease in vitro growth of TSC-associated lung and kidney tumors [35,36,37]. Increased EGF and EGFR expression was found in a TSC mouse model and in tubers and SEGAs of TSC patients [38]. Additionally, increased EGF signaling in TSC brain organoids was linked to the enrichment of a specific interneuron progenitor cell, the caudal late interneuron progenitor (CLIP) [39]. CLIP cells highly express EGFR and possibly are originating cells of both cortical tubers and SEGAs [39]. As organoids consist of both neurons and glial cells, it would be interesting to study whether TSC astrocytes are underlying this increased generation of CLIP-cells or whether it is a cell autonomous effect. Our data, supported by earlier studies, suggests that an increased secretion of EGFR ligands from astrocytes may underlie neuronal pathology in TSC.

The results of the ACM and organoid experiments showed that the synaptic balance in TSC is primed for inhibition, with decreased *VGLUT1* expression and a relative increase in VGAT^+^ synapses. This was a surprising finding, as TSC and epilepsy are generally associated with increased neuronal activity. Indeed TSC neurons are hyperactive in culture [14] and mouse studies have shown decreased synaptic inhibition [17] and reductions in GABA receptor GABA_A_R causing hyperexcitability [16]. It is important to note that VGAT is expressed on GABAergic vesicles located at the presynapse. Although our study showed an increase in number of GABAergic synapses, this does not necessarily correlate to increased inhibitory input at the postsynapse. Indeed, spectroscopy showed increased GABA levels in cortical tubers of TSC patients [40]. A study of *Tsc1*^−/−^ mice showed network hyperexcitability associated with reduced ionotrophic GABAergic receptor per synapse, although an increased number of inhibitory synapses was present, showing compensatory mechanisms may play a role [9]. Also, a recent study using iPSC-derived neuronal monocultures found hyperexcitability together with increased expression of GABAergic genes and decreased expression of *VGLUT1* [15], showing that an increase in GABAergic synapses does not necessarily correlate to increased inhibitory signaling or decreased network signaling. Altogether, while our results are seemingly contradictory to general assumptions, a number of earlier studies have reported increases in GABAergic genes or synapses accompanied by hyperexcitability.

GABA is an inhibitory neurotransmitter in healthy adult brains. However, during development GABA is an excitatory neurotransmitter. The major GABAergic receptor, the GABA_A_ receptor, functions as a chloride channel. During development the intracellular concentration of chloride is higher than the extracellular concentration, causing an efflux of Cl^−^ upon opening chloride channels and a depolarization of the cellular membrane. Waves of GABAergic activity (giant depolarizing potentials; GDPs) in the developing brain drive the formation of glutamatergic synapses and cause an expression shift in chloride transporters which lowers intracellular chloride levels, thus shifting GABA signaling to inhibitory. Changes in GDPs could underlie the decreased *VGLUT1* expression in TSC organoids. Another explanation for *VGLUT1* changes could be a differentiation defect of glutamatergic neurons. Indeed, some previous studies have shown a decreased neuronal generation in TSC [30,41] although this was generally associated with a decreased number of neurons, while the number of NeuN^+^ cells was not changed in our organoids. Other studies did not observe changes in cell type proportions [15,42,43], while changes in neuronal activity and synaptic balance were still observed. It is hypothesized that an incomplete GABAergic shift can cause disruptions of the E/I balance [44]. Expression changes in chloride transporters have been observed in TSC brain tissue [45,46]. A small-scale clinical study showed positive effects of treatment of TSC patients with bumetanide, which can reinstate hyperpolarizing GABA currents [47]. These studies support the hypothesis that GABAergic changes may underlie TSC-associated neurodevelopmental issue like epilepsy and autism.

Previous studies mainly looked at cell autonomous effects in TSC neurons. Interestingly, in our ACM cultures the neurons did not carry TSC mutations, so all changes are mediated through changes in the astrocytic secretome. The data of organoids, in which all cells are derived from either control or TSC, showed the same direction for the *VGAT*/*VGLUT1* ratio, showing that the effects of the astrocyte-secreted factors on synaptic balance are not compensated by changes in TSC neurons. However, it is possible that TSC neurons show additional cell autonomous changes (for example a decrease in GABAergic receptor expression) that will alter the resulting network excitability. Further investigation is necessary to unravel the functional consequences of the relative increase of GABAergic synapses on network excitability.

## 5. Conclusions

To summarize, TSC patient astrocytes are affected and show an increased proliferation and changes in gene expression. Gene expression changes lead to alterations in secreted factors from astrocytes, which in turn influence neuronal development. Secreted factors from TSC astrocytes cause an increase of the percentage of GABAergic (VGAT^+^) synapses, likely altering the balance between GABAergic and glutamatergic synapses. This may be mediated through EGFR signaling as the expression of EGF ligands is increased in TSC astrocytes. The effects of the changes in synaptic balance on neuronal network functioning are currently unclear and should be further investigated. Based on previous studies, it is expected that the changes in GABAergic synapses are accompanied with increased network excitability, either through changes in the GABAergic shift and/or postsynaptic modulations of GABAergic signaling. Astrocytes may provide new therapeutic targets to improve neuronal signaling in TSC patients.

## Figures and Tables

**Figure 1 cells-10-00134-f001:**
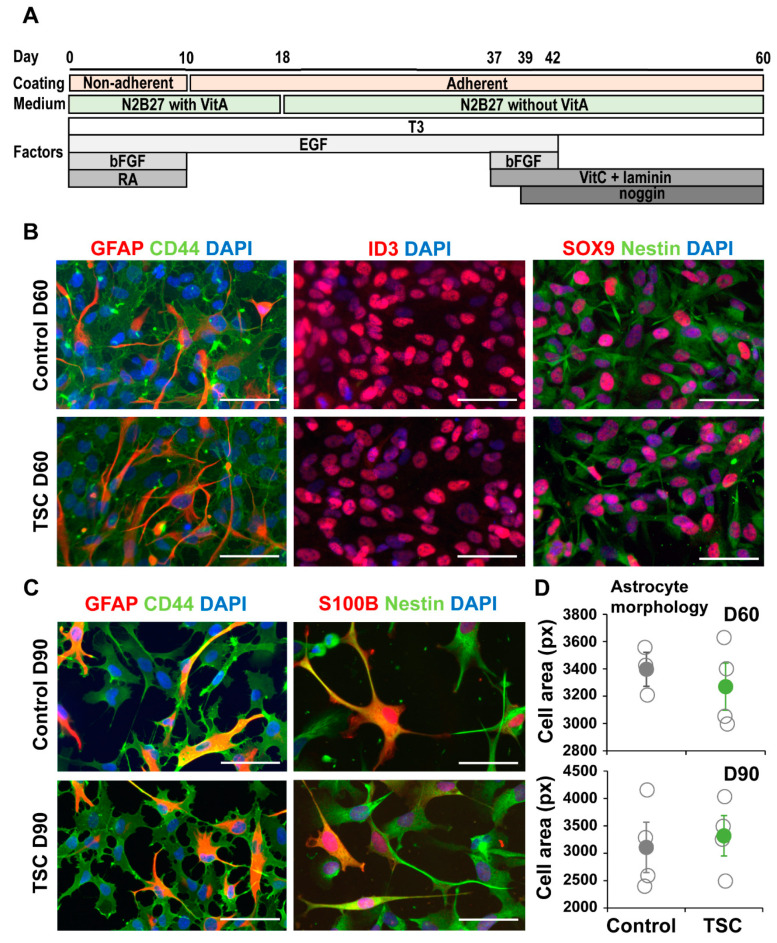
Tuberous sclerosis complex (TSC) iPSCs show normal astrocyte differentiation. (**A**) Schematic overview of the differentiation protocol from iPSC to astrocytes. (**B**) At day 60 of differentiation, cells present with astrocyte morphologies and express many astrocyte-associated markers. Shown are representative stainings of control and TSC astrocytes for astrocytic markers GFAP, CD44, ID3, SOX9 and Nestin. (**C**) Day-90 astrocytes are positive for astrocytic markers like GFAP, CD44, S100B and Nestin. (**D**) Morphological analysis of CD44 staining showed no change in cell size between TSC and control astrocytes at day 60 and day 90. (**B**,**C**) Scalebar = 50 μm. (**D**) Data points represent average cell size per iPSC line, with solid data points representing the mean per condition ± SEM. Statistical test: independent samples *t*-test.

**Figure 2 cells-10-00134-f002:**
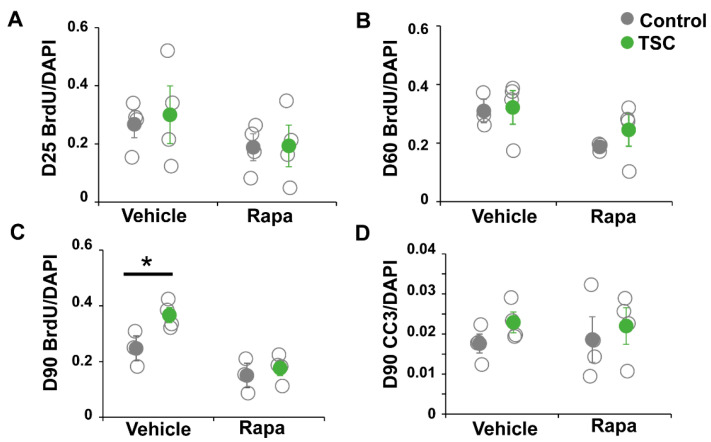
iPSC-derived TSC astrocytes show increased proliferation. A bromodeoxyuridine (BrdU) proliferation assay was performed on day-25 progenitor cells (**A**), day-60 astrocytes (**B**) and day-90 astrocytes (**C**). At day 25 and day 60, no differences in proliferation rate were observed. At day 90, TSC astrocytes showed a significantly increased proliferation compared to control astrocytes, which was rescued by rapamycin treatment. (**D**) No changes were observed in apoptosis rates in TSC cells, measured by cleaved caspase 3 (CC3) staining. (**A**–**D**) Data points represent proliferation (**A**–**C**) or apoptosis (**D**) rates of each iPSC line (average of two experiments) with solid data the points representing mean per condition ± SEM. * = *p* < 0.05 Statistical test: independent samples *t*-test.

**Figure 3 cells-10-00134-f003:**
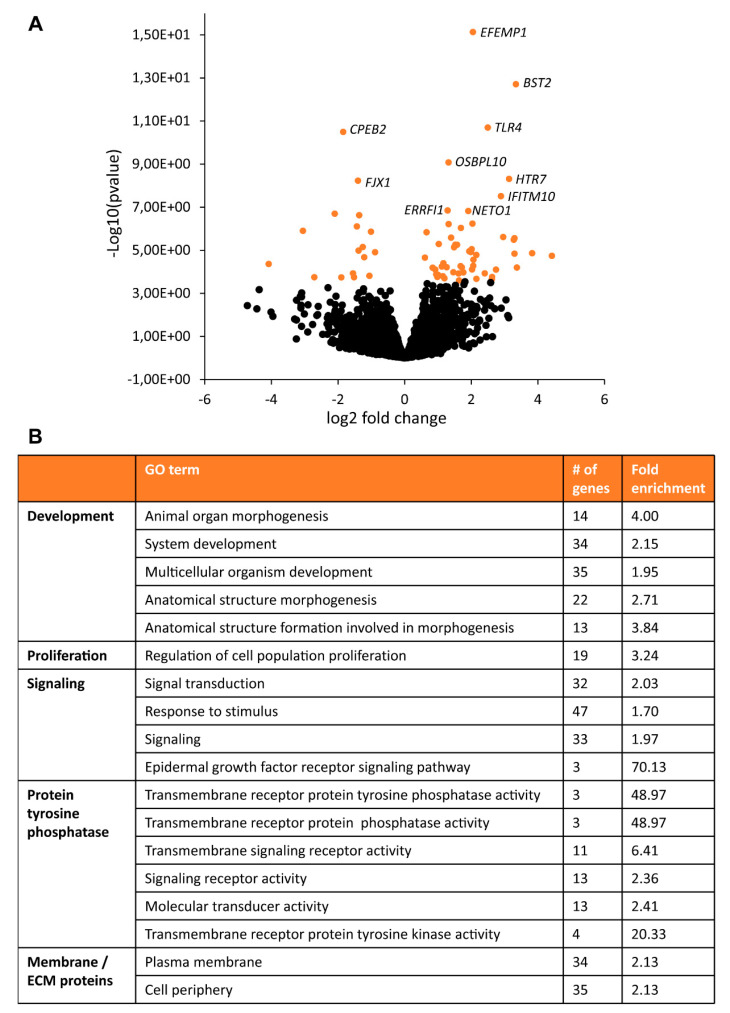
RNA sequencing of TSC astrocytes. RNA sequencing on day-60 control and TSC astrocytes showed 70 differentially expressed genes (DEGs). (**A**) Volcano plot with all significant genes in orange. The top 10 most significant genes are labeled with gene names. (**B**) Pathway enrichment analysis showed a number of enriched GO terms, which could be grouped into the domains “Development”, “Proliferation”, “Signaling”, “Protein tyrosine phosphatase” and “Membrane/ECM proteins”.

**Figure 4 cells-10-00134-f004:**
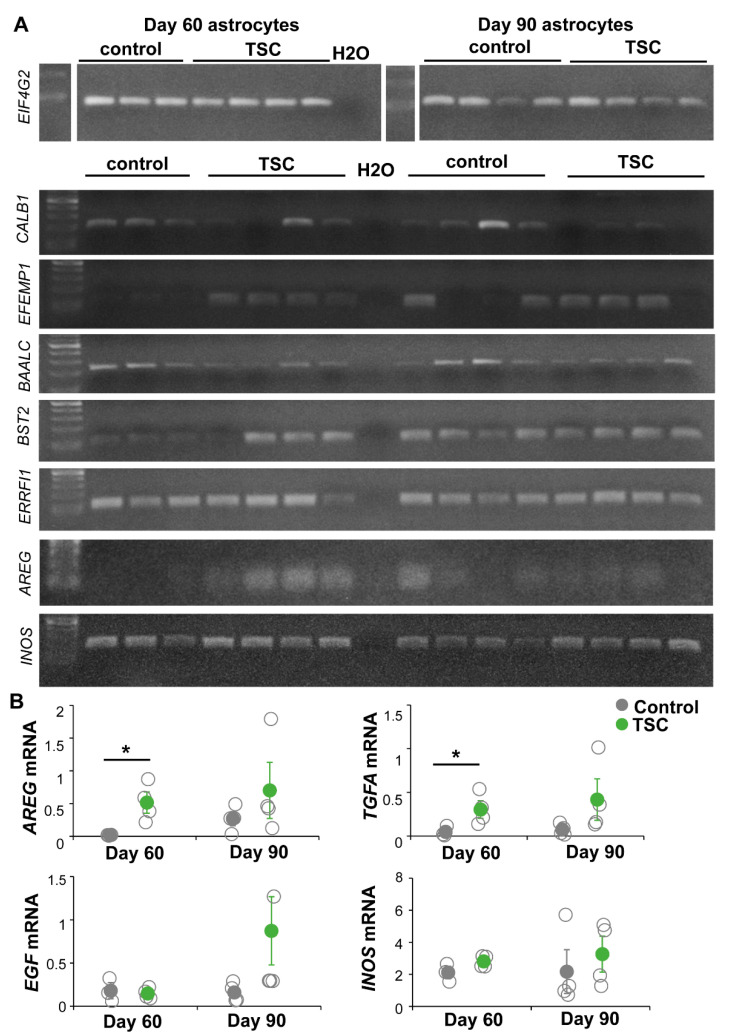
EGF signaling changes are sustained in day-90 astrocytes. The expression of selected DEGs and EGF signaling components are tested in day-60 and day-90 astrocytes. (**A**) PCR products of DEGs *CALB1, EFEMP1, BAALC, BST2, ERRFI1* and *AREG; INOS* which is a downstream target of EGF signaling; and housekeeping gene *EIF4G2.* Most of the differences in day-60 astrocytes are still observed at day 90. (**B**) Bands of EGF signaling components were quantified and corrected for housekeeping gene expression. Significant changes were only observed in *AREG* and *TGFA* in day-60 astrocytes, but all genes show a trend to increased expression in day-90 astrocytes. (**B**) Data points represent data for each iPSC line with solid data points representing the mean per condition ± SEM. * = *p* < 0.05, statistical test: independent samples *t*-test.

**Figure 5 cells-10-00134-f005:**
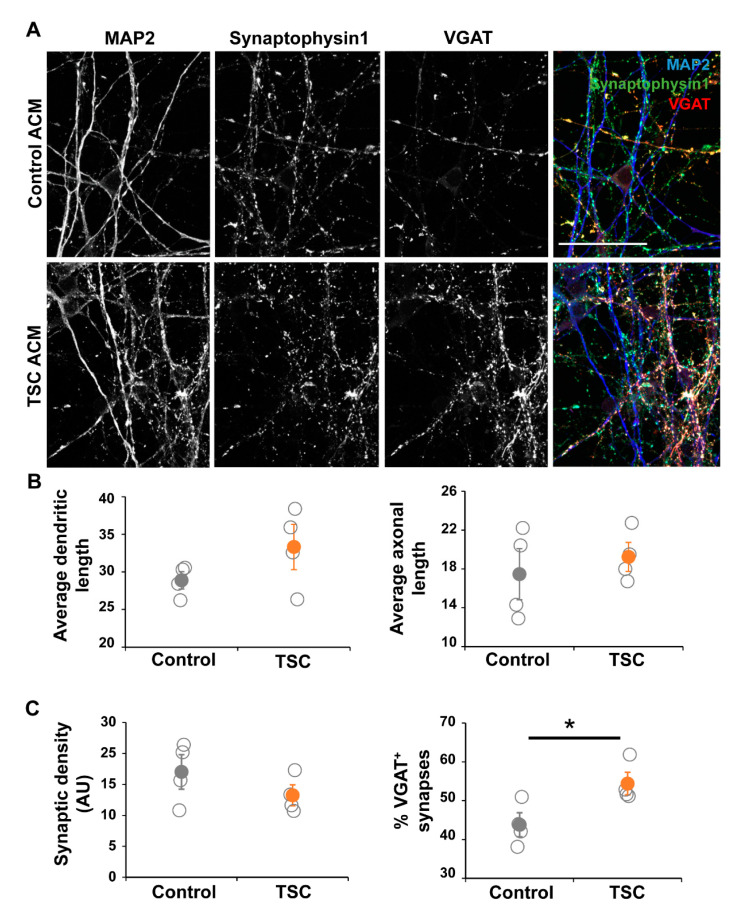
Neurons in TSC ACM show synaptic changes. Control neurons were grown in ACM of either control or TSC astrocytes for 40 days. (**A**) Representative images of immunostaining at the end of the culture (day 57 of neuronal differentiation), showing dendrites by MAP2, all synapses by presynaptic marker Synaptophysin 1, and γ-aminobutyric acid (GABA) synapses by VGAT staining. (**B**) Morphological analysis showed no changes between TSC and control cells. Shown are the results of the average length of dendrites (MAP2^+^ neurites) and axons (SMI312^+^ neurites) per cell. (**C**) Synaptic analysis showed no significant changes in synaptic density, but the percentage of VGAT^+^ synapses was significantly increased in cultures with TSC ACM. ACM = astrocyte-conditioned medium, AU = arbitrary units. (**A**) Confocal images, scale bar = 50 μm (**B**,**C**) data points represent data for each iPSC line (average of two experiments) with solid data points representing the mean per condition ± SEM. * = *p* < 0.05 Statistical test: independent samples *t*-test.

**Figure 6 cells-10-00134-f006:**
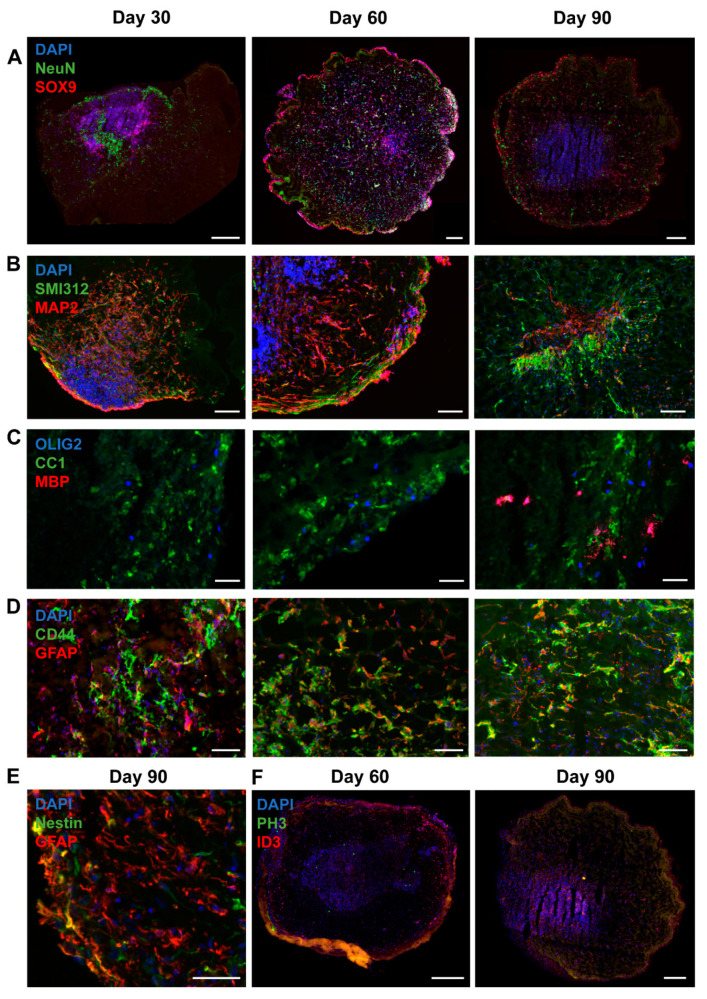
Organoids contain a mix of neural cell populations. Immunostainings at different time points show that a mix of neural cell populations develop in organoids. The presence of neurons is shown by staining for NeuN (**A**), dendritic marker MAP2 and axonal marker SMI312 (**B**). Oligodendrocyte progenitor cells (OLIG2^+^) and oligodendrocytes (CC1^+^) are present at day 30 and day 60, while mature MBP^+^ oligodendrocytes are only observed at day 90 (**C**). Astrocytes were present from day 30 onwards, as shown by staining for SOX9 (**A**), GFAP, CD44 (**D**), Nestin (**E**) and ID3 (**F**). PH3 staining shows proliferating cells in day 60 and day 90 organoids (**F**). (**A**,**F**) Scalebar = 200 μm. (**B**) Scalebar = 100 μm. (**C**–**E**) Scalebar = 50 μm.

**Figure 7 cells-10-00134-f007:**
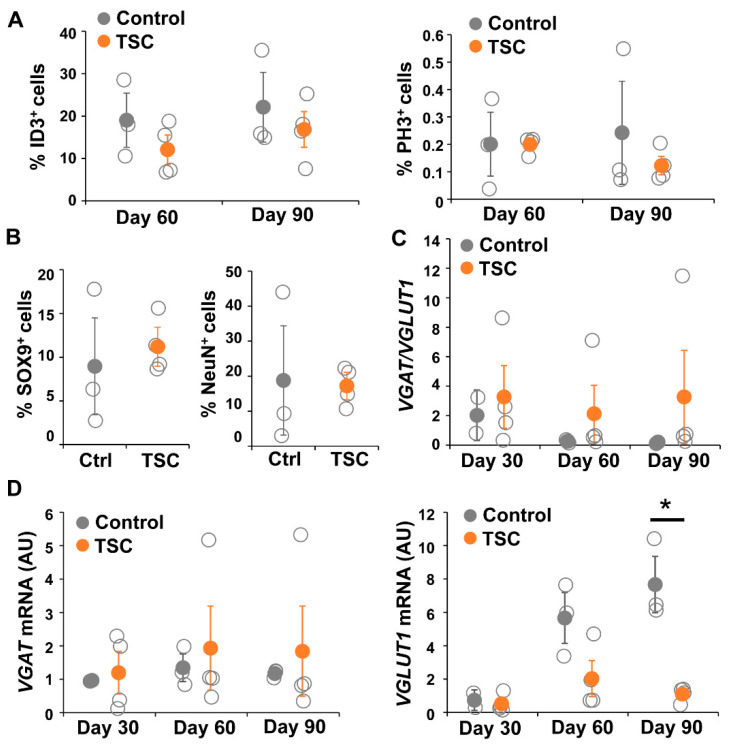
TSC organoids show altered synaptic balance. (**A**) Cell counts of the amount of ID3^+^ cells and PH3^+^ cells showed no differences between TSC and control organoids in the number of astrocytes (ID3) or proliferating cells (PH3). (**B**) Similarly, the percentage of SOX9^+^ and NeuN^+^ cells was not changed between TSC and control organoids at day 90. (**C**,**D**) shows quantitative mRNA analysis for *VGAT* and *VGLUT1* in day 30, day 60 and day 90 TSC and control organoids. *VGLUT1* mRNA is significantly decreased in day 90 TSC organoids compared to controls, leading to an (not-significantly) altered ratio of *VGAT/VGLUT1*. Data points represent data for each iPSC line (average of three or four organoids) with solid data points representing the mean per condition ± SEM. * = *p* < 0.05. Statistical test: independent samples t-test. NC = negative control, AU = arbitrary units.

**Table 1 cells-10-00134-t001:** Overview of induced pluripotent stem cells (iPSC) lines.

Line #	Genotype	Gender	Age	Source
88	Control	M	74 d	Amsterdam UMC, Amsterdam, The Netherlands
228	Control	F	19 y	Amsterdam UMC, Amsterdam, The Netherlands
233	*TSC1*	M	17 y	Coriell Institute, Camden, NJ, USA; GM06149
401	*TSC1*	M	23 y	Coriell Institute, Camden, NJ, USA; GM02332
417	*TSC2*	F	16 y	Coriell Institute, Camden, NJ, USA; GM03958
420	Control	M	21 y	Coriell Institute, Camden, NJ, USA; GM23964
421	Control	M	19 y	Coriell Institute, Camden, NJ, USA; GM23973
424	*TSC2*	F	26 y	Coriell Institute, Camden, NJ, USA; GM06102

**Table 2 cells-10-00134-t002:** Primary antibodies.

Target	Host	Dilution	Company	Number
BrdU	Rabbit	1:500	GeneTex	GTX28091
CC1	Mouse	1:1000	Millipore	OP80
Cleaved Caspase 3	Rabbit	1:400	Cell Signaling	9661
CD44	Mouse	1:100	Hybridomabank	H4C4
GFAP	Rabbit	1:1000	DAKO	Z0334
ID3	Rabbit	1:250	Cell Signaling	9837
MAP2	Chicken	1:5000	Millipore	AB5543
MBP	Rat	1:500	Abcam	Ab7349
Nestin	Mouse	1:1000	BD Bioscience	611658
NeuN	Mouse	1:500	Millipore	MAB377
OLIG2	Rabbit	1:500	Millipore	AB9610
PH3	Mouse	1:1000	Cell Signaling	9706
SOX9	Rabbit	1:500	Cell Signaling	82630
S100B	Rabbit	1:1000	Protein Tech	15146-AP
MAP2	Chicken	1:5000	Millipore	AB5543
SMI312	Mouse	1:1000	Eurogentech	SMI-312P-050
VGAT	Rabbit	1:500	Sysy	131-002
Synaptophysin 1	Guinea Pig	1:1000	Sysy	101-004

**Table 3 cells-10-00134-t003:** Primer sequences.

Target	Forward Primer	Reverse Primer
*AREG*	GATACTCGGCTCAGGCCATT	ATGGTTCACGCTTCCCAGAG
*BAALC*	TGCACTCGGGCTAAAAGAGA	AATTCAGGTCCAGCAAGGGG
*BST2*	GGAAGCTGGCACATCTTGGA	CTAACCGTGTTGCCCCATGA
*CALB1*	GGAAGCATGCCCAAGTGGTATTA	AGCCTTCTTTCGCGCCTGCT
*EFEMP1*	CTCTGCTAGCTCAAGATTCACA	CAGTGCATTGCGTGTACGTG
*EGF*	TCTACTTGTGTGGGTCCTGC	ATCACTGAGACACCAGCATCC
*EIF4G2*	AGGACCGCATGTTGGAGATT	TGAGGGGATGGATCCAACTTT
*ERRFI1*	TGGAGCAGTCGCAGTGAGTTTA	GGAAGCATGCCCAAGTGGTATTA
*INOS*	CGTGGAGACGGGAAAGAAGT	GACCCCAGGCAAGATTTGGA
*NEUN*	TGGCATGACCCTGTACACAC	GCTGCTGCTTCTCTGTAGGG
*TGFA*	TGCCATTCTGGGTACGTTGG	GGACCTGGCAGCAGTGTATC
*VGAT*	GGACTCGTACGTGGCCATAG	AGCTCGATGATCTGCGCTAC
*VGLUT1*	TTCTGGCTGCTCGTCTCCTA	GGTTCATGAGTTTCGCGCTC

## Data Availability

Data is contained within the article or Appendix A.

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
