# Peer review of "Neuron–Glia Interactions in Tuberous Sclerosis Complex Affect the Synaptic Balance in 2D and Organoid Cultures"

_cells, 2021, doi:10.3390/cells10010134_

Round 1
Reviewer 1 Report
This paper aims to uncover astrocytic effects on neurons in TSC, which is important and worthy investigating. The study comprises mainly 2 parts, 1) astrocytes alones, and the effect of their conditoned media on neurons, 2) 3D organoids in which neurons and astrocytes were supposed to development together. The authors noted that 1) ACM from astrcotyes derived from TSC patients produced more VGAT puncta in control neurons. 2) TSC organoids showed a higher VGAT/Vglut1 expressio ratio (qPCR). Based on these two, it is concluded that astrocytes may cause screwed E/I balance which is proposed to be a major contributor to many TSC sympotoms.
While the role of astrocytes in TSC deserves in depth investigations, and i appreciate the study design in general, there are major issues with robustness of the results and other potential interpresentations that prevents the study being accepted at the present form.
1) Organoids data, mainly Fig 4, is not adequate. There are no images to demonstrate the composition (as well as organization and structure) of the brain organoids at all!!! Just qPCR data at a single time point does not mean much. Critically, are there any mature or nearly-mature astrcoytes at d90? Many previous reports have shown that GFAP+ cells do not emerge until much later in brain organoids (>day 150, see pasca et al etc). This cast serious doubt about the effect of astrocytes on neurons, as the authors tried to claim.
2) VGAT/Vglut1 expression ratio, in Fig 4D: i am not convinced abt this. Looking at Fig 4D more carefully on Vglut1 mRNA expression, it is obvious that vglut1 did not increase at all from d30-90 in TSC organoids while it did increase signicantly in WT organoids. Could that be due to the differentiation defects in TSC organoids that it failed to generate excitatory neurons? Authors must show data to address this critical point.
3) The large data variation seen in Fig 4 is problematic. It is known that organoids are subject to lots of variations so more independent batches must be perfomed and data analyzed carefully. Using 3-4 organoids from each line, as shown here, is not acceptable.
4) Fig 1B showed GFAP+ astrocytes at d60. It is known that astrocytes from TSC patient brains are dysmorphilc and hypertrophic. Did the authors observe any of these in their iPSC derived astrocytes? Related to this, Blair et al, 2018 nature medicine paper should be cited and results compared.
5) ACM on neurons is a highlight of the paper. However, only control (wt) neurons were used. Please include data from TSC neurons with ACM. In addition, to confirm the GABA signaling phenotype, inhibitory PSCs (from the neurons in ACM) should be recorded and analyzed.
6) In general, there seems to be some disconnect with first part of results, which is on astrocyte proliferation, and the rest of the data. for example, can rapamycin treatment rescue the VGAT defect? also, given the authors demonstrated a hyperproliferation of astrocytes in 2D, did they observe the same thing in their 3D organoids? More importantly, is this proliferation defect related to the GABA signaling? if so, how? Please explain more.
Author Response
This paper aims to uncover astrocytic effects on neurons in TSC, which is important and worthy investigating. The study comprises mainly 2 parts, 1) astrocytes alones, and the effect of their conditoned media on neurons, 2) 3D organoids in which neurons and astrocytes were supposed to development together. The authors noted that 1) ACM from astrcotyes derived from TSC patients produced more VGAT puncta in control neurons. 2) TSC organoids showed a higher VGAT/Vglut1 expressio ratio (qPCR). Based on these two, it is concluded that astrocytes may cause screwed E/I balance which is proposed to be a major contributor to many TSC sympotoms.
While the role of astrocytes in TSC deserves in depth investigations, and i appreciate the study design in general, there are major issues with robustness of the results and other potential interpresentations that prevents the study being accepted at the present form.
- Organoids data, mainly Fig 4, is not adequate. There are no images to demonstrate the composition (as well as organization and structure) of the brain organoids at all!!! Just qPCR data at a single time point does not mean much. Critically, are there any mature or nearly-mature astrcoytes at d90? Many previous reports have shown that GFAP+ cells do not emerge until much later in brain organoids (>day 150, see pasca et al etc). This cast serious doubt about the effect of astrocytes on neurons, as the authors tried to claim.
Response: We understand the request of the reviewer to show an extensive characterization of the organoids, and therefore included this now in the new Figure 6. Our organoid protocol is an adaptation of the protocol published in Monzel et al., 2017, Stem Cell Reports, where astrocytes were observed from day 60 onwards. In line with this study, our organoids present small number of GFAP+ cells at day 30 (the earliest time point studied) which increase in number by day 60 and day 90 and also express other astrocytic markers like CD44, ID3, SOX9 and Nestin (see Figure 6). In addition to the SOX9 cell counts, we now also included a count of ID3+ cells. We have adjusted the text in the methods section at page 4 line 168-175 and page 5 line 187 (Table 2); to the result section at page 13 line 361-371; to the new Figure 6 on page 14 line 386 and the figure legend on page 14 line 387-394; to Figure 7 on page 15 line 395 and the figure legend on page 15 line 396-398.
- VGAT/Vglut1 expression ratio, in Fig 4D: i am not convinced abt this. Looking at Fig 4D more carefully on Vglut1 mRNA expression, it is obvious that vglut1 did not increase at all from d30-90 in TSC organoids while it did increase signicantly in WT organoids. Could that be due to the differentiation defects in TSC organoids that it failed to generate excitatory neurons? Authors must show data to address this critical point.
Response: Indeed, the differences in synaptic balance in the TSC organoids seem to be caused by VGLUT1 levels that remain low compared to controls. Although much lower than in control, VGLUT1 does increase in TSC organoids over time, with a 4-fold and 2-fold increase, resp. between day 30 and 60 and between day 30 and 90. This shows there is some development of glutamatergic synapses, albeit much lower than in control organoids. Between day 60 and 90, there is actually a decrease in VGLUT1 expression in TSC organoids. Worth noting, this decrease is apparent in all but 1 TSC line (which had a similar VGLUT1 level at day 60 and day 90), while the control lines show a consistent increase in VGLUT1 expression over time. It is possible that there is some differentiation defect of glutamatergic neurons in TSC organoids. Previous studies have shown a differentiation defect in TSC+/- or TSC-/- neurons (Zucco et al., 2018, Mol Cell Neurosci; Blair et al., 2018, Nature Med). However, this was shown by a decrease in the percentage of neuronal cells, while we did not observe changes in NEUN expression or the number of NeuN+ cells. Previous studies using the same iPSC lines did not find changes in cell type proportions in neuronal mono-cultures (Nadadhur et al., 2019, Stem Cell Reports; Alsaqati et al., 2020, Mol. Autism), while a similar decrease in VGLUT1 expression was observed (Alsaqati et al., 2020, Mol. Autism). We have expanded our discussion on the matter on page 16-17 line 470-476.
- The large data variation seen in Fig 4 is problematic. It is known that organoids are subject to lots of variations so more independent batches must be perfomed and data analyzed carefully. Using 3-4 organoids from each line, as shown here, is not acceptable.
Response: It is true that organoids have been reported to have a larger variation than culture products generated via 2D differentiations. Therefore we carefully checked whether data variability we observed was caused by differences between individuals or between organoid structures. Our data shows that the variation observed in the qPCR data is mainly caused by differences between individuals. We have now added this analysis in Figure S1. For all organoid studies we included iPSC lines from 4 TSC patients and 3 control individuals. For each individual we collected at least 2 organoids per time point from at least 2 independent batches. As is shown in Figure S1, the organoids of 1 individual (represented in the same color) actually cluster together, showing that most variation is caused by variation between individuals and not by experimental variation. We have added the new Figure S1, and the figure legend on page 17 line 504-508 and adjusted the text in the result section on page 13 line 378-383.
- Fig 1B showed GFAP+ astrocytes at d60. It is known that astrocytes from TSC patient brains are dysmorphilc and hypertrophic. Did the authors observe any of these in their iPSC derived astrocytes? Related to this, Blair et al, 2018 nature medicine paper should be cited and results compared.
Response: We did not observe differences in morphology of day 60 astrocytes between TSC and control. However, as other abnormalities like proliferation only became apparent in more mature day 90 astrocytes, we decided to now add morphology analysis on day 90 astrocytes. The results showed that also in day 90 astrocytes we did not observe morphological changes in TSC. This is consistent with the results of the Blair et al., 2018 paper. While in the Blair et al., 2018 paper an increased cell size was observed, this was only in homozygous TSC2 knockout cells, and not in the TSC2+/- cells. The patient cells used in our study all carry heterozygous mutations. We have added the data to Figure 1 on page 8 line 285; the figure legend on page 8 line 286-294 and to the results section at page 7 line 275-276. We have extended our discussion on page 15 line 413-415.
- ACM on neurons is a highlight of the paper. However, only control (wt) neurons were used. Please include data from TSC neurons with ACM. In addition, to confirm the GABA signaling phenotype, inhibitory PSCs (from the neurons in ACM) should be recorded and analyzed.
Response: We agree with the reviewer that it would be interesting to study whether cultures with TSC neurons show additional abnormalities. However we feel it is beyond the scope of the current study. The goal of this paper was to show that TSC mutations affect astrocytes, and that these astrocytic changes can influence neuronal development. Extending our study in that direction would need considerable extra time to address these additional questions. Although studying the electrophysiological properties of these neurons may be interesting, we are not sure whether those results will necessarily show more inhibition, as GABA is an excitatory neurotransmitter during neurodevelopment, and alterations in the number of GABAergic synapses may alter the GABAergic shift. To address this point we have now extended the discussion on page 16 line 459-460.
- In general, there seems to be some disconnect with first part of results, which is on astrocyte proliferation, and the rest of the data. for example, can rapamycin treatment rescue the VGAT defect? also, given the authors demonstrated a hyperproliferation of astrocytes in 2D, did they observe the same thing in their 3D organoids? More importantly, is this proliferation defect related to the GABA signaling? if so, how? Please explain more.
Response: The paper can indeed be seen as having two parts; the first part in which we characterized TSC astrocytes, tested whether they showed any known TSC pathology like changes in proliferation, and performed RNA sequencing to assess global changes in TSC astrocytes versus controls. In the second part we took the RNA sequencing data that revealed changes in secreted factors to study the effect of ACM on neurons, and confirmed changes in synaptic balance in an organoid model. Although it would be interesting to see the effect of rapamycin treatment, rapamycin does not only affects astrocytes but also other cell types. Therefore, in complex cultures the cell autonomous effects of rapamycin treatment cannot be easily identified. We have tested proliferation in the organoids by staining for PH3. In day 60 and 90 organoids, we do not see differences in proliferation between TSC and controls. To test whether proliferation is changed specifically in astrocytes, we quantified the number of PH3 and ID3 double positive cells. The number of double positive cells was very low in each organoid, decreasing reliability of the results, but no differences between TSC and control organoids were observed. We included the PH3 data in Figure 7. We do not think that the changes in GABAergic synapses are directly related to increased proliferation in TSC astrocytes, but rather to other changes that were observed in the RNA sequencing like EGF signaling. To further address the role of the EGF pathway, we have included extra analysis of the EGF pathway in day 60 and day 90 astrocytes, to show that indeed EGF signaling is increased in TSC astrocytes. As alterations in EGF signaling are known to effect neuronal development, we think these astrocytic changes may underlie the observed neuronal pathology.
We have added data of proliferation in organoids to Figure 7 on page 15 line 395 and the figure legend on page 15 line 396-398; to the methods section on page 4 line 168-175 and page 5 line 187 (Table 2); to the results section at page 13 line 369-371.
We have added the new data regarding EGF signaling to the methods section on page 6 line 221 and page 6 line 233 (Table 3); to the results section on page 9 line 309-318, to the new Figure 4 on page 11 line 334 and the figure legend on page 11 line 335-343
Reviewer 2 Report
The manuscript by Dooves et al. aims to establish the role of astrocytes differentiated from iPSCs from patients with Tuberous Sclerosis Complex on excitation/inhibition (E/I) balance. At day 60 of astrocytic differentiation, RNA-seq experiments showed alterations in expression of components of signaling pathways, notably the EGF-related transduction. Enhanced proliferation of TSC astrocytes were reported at 90 days of differentiation. Then control iPSCs cells were differentiated to neurons, and the effect of control or TSC astrocyte-conditioned medium (ACM) was used to analyze dendritic and axonal length, as well as the presence of Synaptophysin1 and VGAT. TSC ACM induced an increase in VGAT immuno-labeling. Brain organoids were formed from control and TSC iPSCs. By analyzing transcripts with RT-PCR, markers for neurons, astrocytes and oligodendrocytes were found in organoids of 60 days. In quantitative mRNA assessments, VGLUT1 showed a significant decrease in expression in TSC organoids. VGAT and the VGAT/VGLUT1 ratio did not change at days 30, 60 nor 90.
Major comments:
The conclusions are not supported by the data presented:
- In astrocytic differentiation, changes in RNA-seq were studied at day 60, but the discrete increase in BrdU incorporation was observed at day 90. A selected group of DEG can be used to quantify transcripts at day 90 to see if the changes are long-lasting and can be correlated to the increased proliferation. Additionally, showing the phosphorylation of EGF downstream targets at days 60 and 90 can be informative. Knock-down experiments for the EGF receptor can also help to argument that the increase in proliferation is linked to activation of this receptor.
- The effect of ACM from TSC patients on neuronal differentiation is an increase of approximately 10% on VGAT synapses, when compared to control ACM. However, complementary approaches such as GABA and glutamate depolarization-induced release or electrical recordings with multi-electrodes must be done, to claim an increase in inhibitory transmission caused by TSC ACM.
- The use of organoids as a system to test the participation of astrocytes on the E/I balance is not ideal, since in the TSC organoids neurons can present defects in differentiation. In addition, there are also oligodendrocytes in these structures. In that sense, the ACM might be cleaner, with the combination of the techniques described above. Also, the use of a different technique to determine the presence of VGAT makes the data from the organoids hard to correlate. The presented ratio of mRNA VGAT/mRNA VGLUT1 did no change in the studied time points, making hard to conclude that there is an imbalance towards inhibition.
- In the Conclusions section, the phrase “Secreted factors from TSC astrocytes alter the balance between GABAergic and glutamatergic synapses to a relative increase in GABAergic synapses” is not supported by the experimental data presented.
Minor points:
- In the abstract, “a more physiological model” should be substituted by terms that state that there are neurons and glial cells organized in 3D in organoids.
- The introduction should mention the papers in the cited in Discussion related to decrease in GABAergic responses in TSC.
- The molecular target(s) of rapamycin should be mentioned in Methods and Results.
- The presence of Nestin+ cells at day 60 in Fig. 1B is intriguing. Authors should discuss this and investigate if these cells are present at day 90.
- In Fig. 2B, why are there values of fold enrichment below 2?
- Detection with antibodies in pre-synaptic sites must be performed for VGLUT1 for Fig. 3.
- In Fig. 4A, the band of NEUN is absent. GFAP and VGLUT1 lanes has 2 bands each. The minus RT lane must be included.
- Include pictures of SOX9 and NEUN immuno-detection in organoids in Fig. 4.
Reviewer 3 Report
In this manuscript, Dooves et al report use stem cells derived from patients affected by Tuberous Sclerosis Complex (TSC) to address driving hypothesis that in this disease affected astrocytes are the main cause of excitation/inhibition (E/I) balance in TSC. Authors differentiated stem cells from 4 controls and 4 TSC patients into astrocytes and found their hyperproliferation. In addition, using unbiased transcriptome analysis, authors report that EGF signaling is at the center of transcriptional changes in TSC. Finally, authors used medium form control and TSC astrocytes and found E/I misbalance in neurons when TSC-astrocyte-conditioned media was used. This finding was further confirmed in organoid model. I have only minor concerns:
- Please add name of statistical test used to the figure legends.
- Authors may want to discuss better why would rapamycin rescue the proliferation phenotype observed with TSC astrocytes at 90 days.
- Authors should correct typos in figure legends.
- If image presented taken by a confocal microscope, this should be noted in figure legends.
- Representative images should be included from organoids.
Author Response
In this manuscript, Dooves et al report use stem cells derived from patients affected by Tuberous Sclerosis Complex (TSC) to address driving hypothesis that in this disease affected astrocytes are the main cause of excitation/inhibition (E/I) balance in TSC. Authors differentiated stem cells from 4 controls and 4 TSC patients into astrocytes and found their hyperproliferation. In addition, using unbiased transcriptome analysis, authors report that EGF signaling is at the center of transcriptional changes in TSC. Finally, authors used medium form control and TSC astrocytes and found E/I misbalance in neurons when TSC-astrocyte-conditioned media was used. This finding was further confirmed in organoid model. I have only minor concerns:
- Please add name of statistical test used to the figure legends
Response: We have added the statistical tests to the figure legends on page 8 line 293-294, page 9 line 303, page 11 line 343, page 12 line 356 and page 15 line 404.
- Authors may want to discuss better why would rapamycin rescue the proliferation phenotype observed with TSC astrocytes at 90 days.
Response: Rapamycin is a mTOR inhibitor which could therefore directly compensate the decreased mTOR inhibition observed in TSC cells. High levels of mTOR signaling are known to increase proliferation and therefore rapamycin is able to decrease proliferation directly. Interestingly, it can be observed that rapamycin also slightly decreases proliferation in control cells, showing this is not a TSC specific effect. We have added this information to the discussion on page 15 line 415-417. Additionally, we added more information about the molecular target of rapamycin in the methods section on page 5 line 191-192, and in the results section on page 8 line 281-282.
- Authors should correct typos in figure legends.
Response: We apologize for the oversight of some mistakes and corrected the figure legends.
- If image presented taken by a confocal microscope, this should be noted in figure legends.
Response: We have added in the figure legends if images were taken by confocal microscopy, which only applied to the stainings in Figure 5 on page 12 line 354.
- Representative images should be included from organoids.
Response: We have added representative images from the organoids showing stainings for astrocytes, neurons and oligodendrocytes. We have adjusted the text in the methods section at page 4 line 168-175 and page 5 line 187 (Table 2); to the result section at page 13 line 361-371; to the new Figure 6 on page 14 line 386 and the figure legend on page 14 line 387-394.
Round 2
Reviewer 2 Report
The authors made subtle modifications to the manuscript, including new data on RT-PCR in astrocytes at D90; three out of five DEG analyzed preserve significant differences. Regarding the expression of EGF-related genes AREG, TGFA, EGF and iNOS, there are no significant differences between control and TSC at D90. Furthermore, the presence of transcripts is not enough to show activation of these receptors. Increased phosphorylation of proteins by western blot is the appropriate experiment to show. Additionally, the authors now show staining of control organoids in Fig. 6 without significant differences in any of the markers between control and TSC cells in Fig. 7. But the most important parts of the previous comments were not addressed, namely, no change in the E/I is shown. The authors insist that the non-significant change in the changes in transcripts for VGAT/VGLUT1 is enough to conclude an increase in inhibition in TSC cells. Either the measurement of GABA and glutamate by HPLC after depolarization, or electrical recordings of spontaneous activity in the 2D or 3D systems, are a must to support the title of the manuscript. In their response, the authors accept this fact: “It is true that we cannot conclude an increase in inhibitory transmission by TSC ACM”. This is true for the organoids as well. Therefore, I still consider that the presented data is not enough for publication.
Author Response
Reviewer 2:
The authors made subtle modifications to the manuscript, including new data on RT-PCR in astrocytes at D90; three out of five DEG analyzed preserve significant differences. Regarding the expression of EGF-related genes AREG, TGFA, EGF and iNOS, there are no significant differences between control and TSC at D90. Furthermore, the presence of transcripts is not enough to show activation of these receptors. Increased phosphorylation of proteins by western blot is the appropriate experiment to show. Additionally, the authors now show staining of control organoids in Fig. 6 without significant differences in any of the markers between control and TSC cells in Fig. 7. But the most important parts of the previous comments were not addressed, namely, no change in the E/I is shown. The authors insist that the non-significant change in the changes in transcripts for VGAT/VGLUT1 is enough to conclude an increase in inhibition in TSC cells. Either the measurement of GABA and glutamate by HPLC after depolarization, or electrical recordings of spontaneous activity in the 2D or 3D systems, are a must to support the title of the manuscript. In their response, the authors accept this fact: “It is true that we cannot conclude an increase in inhibitory transmission by TSC ACM”. This is true for the organoids as well. Therefore, I still consider that the presented data is not enough for publication.
Response: We conclude a change in the E/I balance based on 1) the increased VGAT+ synapses in neurons cultured in TSC ACM, and 2) the decreased VGLUT1 expression in TSC organoids. Both findings are based on statistical significant differences between TSC and control cells. We additionally mention that the VGAT/VGLUT1 ratio in organoids is higher in all TSC lines compared to control lines. This finding indeed does not reach statistical significance (p=0.077), but can be explained by high variability in TSC lines (the difference is statistically significant if we leave out the line with the highest VGAT/VGLUT1 ratio). We agree with the reviewer that the changes in VGAT/VGLUT1 ratio were presented too strong in some parts of the manuscript. We have now made alterations in the text to better represent the results.
Our results show a change in the amount of GABAergic and glutamatergic synapses in TSC neurons. This means that the results show changes in E/I balance on a synaptic level. So indeed, we cannot conclude an increase in inhibitory transmission as during development GABAergic synapses are not always inhibitory. Although we agree with the reviewer that it would be interesting to study how these synaptic changes correlate to changes in signal transmission, we feel this is beyond the scope of the current paper in which we aimed to show that secreted factors from astrocytes in TSC affect neuronal development.
We have altered the introduction on page 2 line 70, and the subtitle of the result section on organoids on page 13 line 358. We have clarified that the VGAT/VGLUT1 ratio in organoids is not a statistically significant change on page 16 line 425-427 in the discussion. We have clarified that we are talking about a change in synaptic E/I balance in the title on page 1 line 3,the result section on page 15 line 396 and in the conclusion on page 17 line 497-498.